# DASpeech: Directed Acyclic Transformer for Fast and High-quality Speech-to-Speech Translation

**Qingkai Fang**[1,2], **Yan Zhou**[1,2], **Yang Feng**[1,2*]
[1]Key Laboratory of Intelligent Information Processing
Institute of Computing Technology, Chinese Academy of Sciences (ICT/CAS)
[2]University of Chinese Academy of Sciences, Beijing, China
{fangqingkai21b,zhouyan23z,fengyang}@ict.ac.cn

## Abstract

Direct speech-to-speech translation (S2ST) translates speech from one language into another using a single model. However, due to the presence of linguistic and acoustic diversity, the target speech follows a complex multimodal distribution, posing challenges to achieving both high-quality translations and fast decoding speeds for S2ST models. In this paper, we propose DASpeech, a non-autoregressive direct S2ST model which realizes both *fast* and *high-quality* S2ST. To better capture the complex distribution of the target speech, DASpeech adopts the two-pass architecture to decompose the generation process into two steps, where a linguistic decoder first generates the target text, and an acoustic decoder then generates the target speech based on the hidden states of the linguistic decoder. Specifically, we use the decoder of DA-Transformer as the linguistic decoder, and use FastSpeech 2 as the acoustic decoder. DA-Transformer models translations with a directed acyclic graph (DAG). To consider all potential paths in the DAG during training, we calculate the expected hidden states for each target token via dynamic programming, and feed them into the acoustic decoder to predict the target mel-spectrogram. During inference, we select the most probable path and take hidden states on that path as input to the acoustic decoder. Experiments on the CVSS Fr→En benchmark demonstrate that DASpeech can achieve comparable or even better performance than the state-of-the-art S2ST model Translatotron 2, while preserving up to $18.53\times$ speedup compared to the autoregressive baseline. Compared with the previous non-autoregressive S2ST model, DASpeech does not rely on knowledge distillation and iterative decoding, achieving significant improvements in both translation quality and decoding speed. Furthermore, DASpeech shows the ability to preserve the speaker's voice of the source speech during translation.[2][3]

## 1 Introduction

Direct speech-to-speech translation (S2ST) directly translates speech of the source language into the target language, which can break the communication barriers between different language groups and has broad application prospects. Traditional S2ST usually consists of cascaded automatic speech recognition (ASR), machine translation (MT), and text-to-speech (TTS) models [1, 2]. In contrast, direct S2ST achieves source-to-target speech conversion with a unified model [3], which can (1) avoid error propagation across sub-models [4]; (2) reduce the decoding latency [5]; and (3) preserve

---

[*]Corresponding author: Yang Feng.

[2]Audio samples are available at https://ictnlp.github.io/daspeech-demo/.
[3]Code is publicly available at https://github.com/ictnlp/DASpeech.

non-linguistic information (e.g., the speaker's voice) during translation [6]. Recent works show that direct S2ST can achieve comparable or even better performance than cascaded systems [7, 8].

Despite the theoretical advantages of direct S2ST, it is still very challenging to train a direct S2ST model in practice. Due to the linguistic diversity during translation, as well as the diverse acoustic variations (e.g., duration, pitch, energy, etc.), the target speech follows a complex multimodal distribution. To address this issue, Jia et al. [6], Inaguma et al. [7] propose the two-pass architecture, which first generates the target text with a linguistic decoder, and then uses an acoustic decoder to generate the target speech based on the hidden states of the linguistic decoder. The two-pass architecture decomposes the generation process into two steps: content translation and speech synthesis, making it easier to model the complex distribution of the target speech and achieving state-of-the-art performance among direct S2ST models.

Although the two-pass architecture achieves better translation quality, two passes of autoregressive decoding also incur high decoding latency. To reduce the decoding latency, Huang et al. [9] recently proposes non-autoregressive (NAR) S2ST that generates target speech in parallel. However, due to the conditional independence assumption of NAR models, it becomes more difficult to capture the multimodal distribution of the target speech compared with autoregressive models[4]. Therefore, the trade-off between translation quality and decoding speed of S2ST remains a pressing issue.

In this paper, we introduce an S2ST model with both high-quality translations and fast decoding speeds: DASpeech, a non-autoregressive two-pass direct S2ST model. Like previous two-pass models, DASpeech includes a speech encoder, a linguistic decoder, and an acoustic decoder. Specifically, the linguistic decoder uses the structure of DA-Transformer [11] decoder, which models translations via a directed acyclic graph (DAG). The acoustic decoder adopts the design of FastSpeech 2 [12], which takes the hidden states of the linguistic decoder as input and generates the target mel-spectrogram. During training, we consider all possible paths in the DAG by calculating the expected hidden state for each target token via dynamic programming, which are fed to the acoustic decoder to predict the target mel-spectrogram. During inference, we first find the most probable path in DAG and take hidden states on that path as input to the acoustic decoder. Due to the task decomposition of two-pass architecture, as well as the ability of DA-Transformer and FastSpeech 2 themselves to model linguistic diversity and acoustic diversity, DASpeech is able to capture the multimodal distribution of the target speech. Experiments on the CVSS Fr→En benchmark show that: (1) DASpeech achieves comparable or even better performance than the state-of-the-art S2ST model Translatotron 2, while maintaining up to $18.53\times$ speedup compared to the autoregressive model. (2) Compared with the previous NAR S2ST model TranSpeech [9], DASpeech no longer relies on knowledge distillation and iterative decoding, achieving significant advantages in both translation quality and decoding speed. (3) When training on speech-to-speech translation pairs of the same speaker, DASpeech emerges with the ability to preserve the source speaker's voice during translation.

## 2 Background

### 2.1 Directed Acyclic Transformer

Directed Acyclic Transformer (DA-Transformer) [11, 13] is proposed for non-autoregressive machine translation (NAT), which achieves comparable results to autoregressive Transformer [14] without relying on knowledge distillation. DA-Transformer consists of a Transformer encoder and an NAT decoder. The hidden states of the last decoder layer are organized as a directed acyclic graph (DAG). The hidden states correspond to vertices of the DAG, and there are unidirectional edges that connect vertices with small indices to those with large indices. DA-Transformer successfully alleviates the linguistic multi-modality problem since DAG can capture multiple translations simultaneously by assigning different translations to different paths in DAG.

Formally, given a source sequence $X = (x_1, ..., x_N)$ and a target sequence $Y = (y_1, ..., y_M)$, the encoder takes $X$ as input and the decoder takes learnable positional embeddings $\mathbf{G} = (\mathbf{g}_1, ..., \mathbf{g}_L)$ as input. Here $L$ is the graph size, which is set to $\lambda$ times the source length, i.e., $L = \lambda \cdot N$, and $\lambda$ is a hyperparameter. DA-Transformer models the *translation probability* $P_\theta(Y|X)$ by marginalizing all

---

[4]It is known as the multi-modality problem [10] in non-autoregressive sequence generation.

possible paths in DAG:

$$P_\theta(Y|X) = \sum_{A \in \Gamma} P_\theta(Y|A, X)P_\theta(A|X), \tag{1}$$

where $A = (a_1, ..., a_M)$ is a path represented by a sequence of vertex indexes with $1 = a_1 < \cdots < a_M = L$, and $\Gamma$ contains all paths with the same length as the target sequence $Y$. The probability of path $A$ is defined as:

$$P_\theta(A|X) = \prod_{i=1}^{M-1} P_\theta(a_{i+1}|a_i, X) = \prod_{i=1}^{M-1} \mathbf{E}_{a_i, a_{i+1}}, \tag{2}$$

where $\mathbf{E} \in \mathbb{R}^{L \times L}$ is the *transition probability matrix*. We apply lower triangular masking on $\mathbf{E}$ to allow only forward transitions. With the selected path $A$, all target tokens are predicted in parallel:

$$P_\theta(Y|A, X) = \prod_{i=1}^{M} P_\theta(y_i|a_i, X) = \prod_{i=1}^{M} \mathbf{P}_{a_i, y_i}, \tag{3}$$

where $\mathbf{P} \in \mathbb{R}^{L \times |\mathbb{V}|}$ is the *prediction probability matrix*, and $\mathbb{V}$ indicates the vocabulary. Finally, we train the DA-Transformer by minimizing the negative log-likelihood loss:

$$\mathcal{L}_{\text{DAT}} = -\log P_\theta(Y|X) = -\log \sum_{A \in \Gamma} P_\theta(Y|A, X)P_\theta(A|X), \tag{4}$$

which can be calculated with dynamic programming.

## 2.2 FastSpeech 2

FastSpeech 2 [12] is a non-autoregressive text-to-speech (TTS) model that generates mel-spectrograms from input phoneme sequences in parallel. It is composed of three stacked modules: encoder, variance adaptor, and mel-spectrogram decoder. The encoder and mel-spectrogram decoder consist of several feed-forward Transformer blocks, each containing a self-attention layer followed by a 1D-convolutional layer. The variance adaptor contains three variance predictors including duration predictor, pitch predictor, and energy predictor, which are used to reduce the information gap between input phoneme sequences and output mel-spectrograms. During training, the ground truth duration, pitch and energy are used to train these variance predictors and also as conditional inputs to generate the mel-spectrogram. During inference, we use the predicted values of these variance predictors. The introduction of variation information greatly alleviates the acoustic multi-modality problem, which leads to better voice quality. The training objective of FastSpeech 2 consists of four terms:

$$\mathcal{L}_{\text{TTS}} = \mathcal{L}_{\text{L1}} + \mathcal{L}_{\text{dur}} + \mathcal{L}_{\text{pitch}} + \mathcal{L}_{\text{energy}}, \tag{5}$$

where $\mathcal{L}_{\text{L1}}$ measures the L1 distance between the predicted and ground truth mel-spectrograms, $\mathcal{L}_{\text{dur}}$, $\mathcal{L}_{\text{pitch}}$ and $\mathcal{L}_{\text{energy}}$ compute the mean square error (MSE) loss between predictions and ground truth for duration, pitch, and energy, respectively.

## 3 DASpeech

In this section, we introduce DASpeech, a non-autoregressive two-pass direct S2ST model that generates target phonemes and target mel-spectrograms successively. Formally, the source speech sequence is denoted as $X = (x_1, ..., x_N)$, where $N$ is the number of frames in the source speech. The sequences of target phoneme and target mel-spectrogram are represented by $Y = (y_1, ..., y_M)$ and $S = (s_1, ..., s_T)$, respectively. DASpeech first generates $Y$ from $X$ with a speech-to-text translation (S2TT)[5] DA-Transformer. Subsequently, it generates $S$ with a FastSpeech 2-style decoder conditioned on the last-layer hidden states of the DA-Transformer. We first overview the model architecture of DASpeech in Section 3.1. In Section 3.2, we introduce our proposed training techniques that leverage pretrained S2TT DA-Transformer and FastSpeech 2 models and finetune the entire model for S2ST end-to-end. Finally, we present several decoding algorithms for DASpeech in Section 3.3.

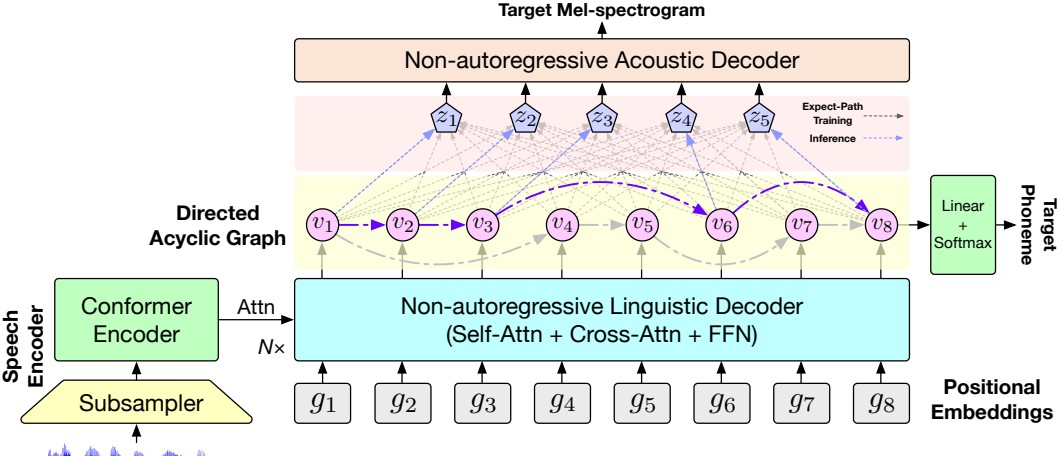

Figure 1: Overview of DASpeech. The last-layer hidden states of the linguistic decoder are organized as a DAG. During training, the input to the acoustic decoder is the sequence of expected hidden states. During inference, it is the sequence of hidden states on the most probable path.

## 3.1 Model Architecture

As shown in Figure 1, DASpeech consists of three parts: a speech encoder, a non-autoregressive linguistic decoder, and a non-autoregressive acoustic decoder. Below are the details of each part.

**Speech Encoder** Since our model takes speech features as input, we replace the Transformer encoder in the original DA-Transformer with a speech encoder. The speech encoder contains a subsampler followed by several Conformer blocks [15]. Specifically, the subsampler consists of two 1D-convolutional layers which shrink the length of input sequences by a factor of 4. Conformer combines multi-head attention modules and convolutional layers together to capture both global and local features. We use relative positional encoding [16] in the multi-head attention module.

**Non-autoregressive Linguistic Decoder** The linguistic decoder is identical to the decoder of DA-Transformer, which generates the target phoneme sequence from the source speech in parallel. Each decoder layer comprises a self-attention layer, a cross-attention layer, and a feed-forward layer. The decoder takes learnable positional embeddings as input, and the last-layer hidden states are organized as a DAG to model translations, as we described in Section 2.1.

**Non-autoregressive Acoustic Decoder** The acoustic decoder adopts the design of FastSpeech 2, which generates target mel-spectrograms from the last-layer hidden states of DA-Transformer in parallel. The model architecture is the same as that introduced in Section 2.2, except that the embedding matrix of the input phonemes is removed, since the input has changed from the phoneme sequence to the hidden state sequence.

## 3.2 Training

DASpeech has the advantage of easily utilizing pretrained S2TT DA-Transformer and FastSpeech 2 models. We use the pretrained S2TT DA-Transformer to initialize the speech encoder and the non-autoregressive linguistic decoder, and use the pretrained FastSpeech 2 model to initialize the non-autoregressive acoustic decoder. Finally, the entire model is finetuned for direct S2ST. This pretraining-finetuning pipeline simplifies model training and enables the use of additional S2TT and TTS data. However, end-to-end finetuning presents a major challenge due to the length discrepancy between the output from the linguistic decoder and the input to the acoustic decoder. Specifically, the hidden state sequence output by the linguistic decoder has a length of $L = \lambda \cdot N$, while the input sequence required by the acoustic decoder should have a length of $M$, which is the length of the ground truth phoneme sequence. Therefore, it is necessary to determine how to obtain the input

---

[5]In this work, speech-to-text translation refers to speech-to-phoneme translation if not otherwise specified.

sequence of acoustic decoder $\mathbf{Z} = (\mathbf{z}_1, ..., \mathbf{z}_M)$ from the last-layer hidden states of linguistic decoder $\mathbf{V} = (\mathbf{v}_1, ..., \mathbf{v}_L)$. The following introduces our proposed approach: *Expect-Path Training*.

**Expect-Path Training** Intuitively, the $i$-th input element $\mathbf{z}_i$ should be the hidden state of the vertex responsible for generating $y_i$. However, since there may be multiple vertices capable of generating each $y_i$ due to numerous possible paths, we would like to consider all potential paths. To address this issue, we define $\mathbf{z}_i$ as the expected hidden state under the posterior distribution $P_\theta(a_i|X, Y)$:

$$\mathbf{z}_i = \sum_{j=1}^{L} P_\theta(a_i = j|X, Y) \cdot \mathbf{v}_j, \tag{6}$$

where $P_\theta(a_i = j|X, Y)$ refers to the probability of vertex $j$ being the $i$-th vertex on path $A$, which means that $y_i$ is generated by vertex $j$. We can compute $P_\theta(a_i = j|X, Y)$ as follows:

$$P_\theta(a_i = j|X, Y) = \sum_{A \in \Gamma} \mathbb{1}(a_i = j) \cdot P_\theta(A|X, Y) \tag{7}$$

$$= \sum_{A \in \Gamma} \mathbb{1}(a_i = j) \cdot \frac{P_\theta(Y, A|X)}{\sum_{A' \in \Gamma} P_\theta(Y, A'|X)} \tag{8}$$

$$= \frac{\sum_{A \in \Gamma} \mathbb{1}(a_i = j) \cdot P_\theta(Y, A|X)}{\sum_{A \in \Gamma} P_\theta(Y, A|X)}, \tag{9}$$

where $\mathbb{1}(a_i = j)$ is an indicator function to indicate whether the $i$-th vertex of path $A$ is vertex $j$. To calculate $\sum_{A \in \Gamma} \mathbb{1}(a_i = j) \cdot P_\theta(Y, A|X)$ and $\sum_{A \in \Gamma} P_\theta(Y, A|X)$ in Equation (9), we employ the *forward-backward algorithm* [17], which involves two passes of dynamic programming.

***Forward Algorithm*** The *forward probability* is defined as $\alpha_i(j) = P_\theta(y_1, ..., y_i, a_i = j|X)$, which is the probability of generating the partial target sequence $(y_1, ..., y_i)$ and ending in vertex $j$ at the $i$-th step. By definition, we have $\alpha_1(1) = \mathbf{P}_{1,y_1}$ and $\alpha_1(1 < j \leq L) = 0$. Due to the Markov property, we can sequentially calculate $\alpha_i(\cdot)$ from its previous step $\alpha_{i-1}(\cdot)$ as follows:

$$\alpha_i(j) = \mathbf{P}_{j,y_i} \sum_{k=1}^{j-1} \alpha_{i-1}(k) \cdot \mathbf{E}_{k,j}. \tag{10}$$

***Backward Algorithm*** The *backward probability* is defined as $\beta_i(j) = P_\theta(y_{i+1}, ..., y_M|a_i = j, X)$, which is the probability of starting from vertex $j$ at the $i$-th step and generating the rest of the target sequence $(y_{i+1}, ..., y_M)$. By definition, we have $\beta_M(L) = 1$ and $\beta_M(1 \leq j < L) = 0$. Similar to the forward algorithm, we can sequentially calculate $\beta_i(j)$ from its next step $\beta_{i+1}(j)$ as follows:

$$\beta_i(j) = \sum_{k=j+1}^{L} \mathbf{E}_{j,k} \cdot \beta_{i+1}(k) \cdot \mathbf{P}_{k,y_{i+1}}. \tag{11}$$

Recalling Equation (9), the denominator is the sum of the probabilities of all valid paths, which is equal to $\alpha_M(L)$. The numerator is the sum of the probabilities of all paths with $a_i = j$, which is equal to $\alpha_i(j) \cdot \beta_i(j)$. Therefore, the Equation (6) can be calculated as:

$$\mathbf{z}_i = \sum_{j=1}^{L} P_\theta(a_i = j|X, Y) \cdot \mathbf{v}_j = \sum_{j=1}^{L} \frac{\alpha_i(j) \cdot \beta_i(j)}{\alpha_M(L)} \cdot \mathbf{v}_j. \tag{12}$$

The time complexity of the forward-backward algorithm is $\mathcal{O}(ML^2)$. Finally, the training objective of DASpeech is as follows:

$$\mathcal{L}_{\text{DASpeech}} = \mathcal{L}_{\text{DAT}} + \mu \cdot \mathcal{L}_{\text{TTS}}, \tag{13}$$

where $\mu$ is the weight of TTS loss. The definitions of $\mathcal{L}_{\text{DAT}}$ and $\mathcal{L}_{\text{TTS}}$ are the same as those in Equations (4) and (5).

## 3.3 Inference

During inference, we perform two-pass parallel decoding. First, we find the most probable path $A^*$ in DAG with one of the decoding strategies proposed for DA-Transformer (see details below). We then

feed the last-layer hidden states on path $A^*$ to the non-autoregressive acoustic decoder to generate the mel-spectrogram. Finally, the predicted mel-spectrogram will be converted into waveform using a pretrained HiFi-GAN vocoder [18]. Since both DAG and TTS decoding are fully parallel, DASpeech achieves significant improvements in decoding efficiency compared to previous two-pass models which rely on two passes of autoregressive decoding. Considering the trade-off between translation quality and decoding efficiency, we use the following two decoding strategies for DA-Transformer in our experiments: *Lookahead* and *Joint-Viterbi*.

**Lookahead** Lookahead decoding sequentially chooses $a_i$ and $y_i$ in a greedy way. At each decoding step, it jointly considers the transition probability and the prediction probability:

$$a_i^*, y_i^* = \arg\max_{a_i, y_i} P_\theta(y_i|a_i, X)P_\theta(a_i|a_{i-1}, X).$$ (14)

**Joint-Viterbi** Joint-Viterbi decoding [19] finds the global joint optimal solution of the translation and decoding path via Viterbi decoding [20]:

$$A^*, Y^* = \arg\max_{A, Y} P_\theta(Y, A|X).$$ (15)

After Viterbi decoding, we first decide the target length $M$ and obtain the optimal path by backtracking from $a_M^* = L$. More details can be found in the original paper.

## 4 Experiments

### 4.1 Experimental Setup

**Dataset** We conduct experiments on the CVSS dataset [4], a large-scale S2ST corpus containing speech-to-speech translation pairs from 21 languages to English. It is extended from the CoVoST 2 [21] S2TT corpus by synthesizing the target text into speech with state-of-the-art TTS models. It includes two versions: CVSS-C and CVSS-T. For CVSS-C, all target speeches are in a single speaker's voice. For CVSS-T, the target speeches are in voices transferred from the corresponding source speeches. We evaluate the models on the CVSS-C French→English (Fr→En) and CVSS-T French→English (Fr→En) datasets. We also conduct a multilingual experiment by combining all 21 language directions in CVSS-C together to train a single many-to-English S2ST model.

**Pre-processing** We convert the source speech to 16000Hz and generate target speech with 22050Hz. We compute the 80-dimensional mel-filterbank features for the source speech, and transform the target waveform into mel-spectrograms following Ren et al. [12]. We apply utterance-level and global-level cepstral mean-variance normalization for source speech and target speech, respectively. We follow Ren et al. [12] to extract the duration, pitch, and energy information of the target speech.

**Model Configurations** The speech encoder, linguistic decoder, and acoustic decoder contain 12 Conformer layers, 4 Transformer decoder layers, and 8 feed-forward Transformer blocks, respectively. The detailed configurations can be found in Table 5 in Appendix A. For model regularization, we set dropout to 0.1 and weight decay to 0.01, and no label smoothing is used. We use the HiFi-GAN vocoder pretrained on the VCTK dataset[6] [22] to convert the mel-spectrogram into waveform.

**Training** DASpeech follows the pretraining-finetuning pipeline. During pretraining, the speech encoder and the linguistic decoder are trained on the S2TT task for 100k updates with a batch of 320k audio frames. The learning rate warms up to 5e-4 within 10k steps. The acoustic decoder is pretrained on the TTS task for 100k updates with a batch size of 512. The learning rate warms up to 5e-4 within 4k steps. During finetuning, we train the entire model for 50k updates with a batch of 320k audio frames. The learning rate warms up to 1e-3 within 4k steps. We use Adam optimizer [23] for both pretraining and finetuning. For the weight of TTS loss $\mu$, we experiment with $\mu \in \{1.0, 2.0, 5.0, 10.0\}$ and choose $\mu = 5.0$ according to results on the `dev` set. We implement our model with the open-source toolkit *fairseq* [24]. All models are trained on 4 RTX 3090 GPUs.

In the multilingual experiment, the presence of languages with limited data or substantial interlingual variations makes the mapping from source speech to target phonemes particularly challenging. To address this, we adopt a two-stage pretraining strategy. Initially, we pretrain the speech encoder and

---

[6]See VCTK_V1 in `https://github.com/jik876/hifi-gan`.

Table 1: Results on CVSS-C Fr→En and CVSS-T Fr→En `test` sets. ♣ indicates results quoted from Huang et al. [9]. ♠ indicates results of our re-implementation. †: target length beam=15 and noisy parallel decoding (NPD). $T_{\text{phone}}$, $T_{\text{unit}}$, and $T_{\text{mel}}$ indicate the sequence length of phonemes, discrete units, and mel-spectrograms, respectively. ** means the improvements over S2UT are statistically significant ($p < 0.01$).

| ID | Models | Decoding | #Iter | #Param | ASR-BLEU (Fr→En) CVSS-C | CVSS-T | Speedup |
|---|---|---|---|---|---|---|---|
| | Ground Truth | / | / | / | 84.52 | 81.48 | / |
| *Single-pass autoregressive decoding* | | | | | | | |
| A1 | S2UT [5] | Beam=10 | $T_{\text{unit}}$ | 73M | 22.23 | 22.28 | 1.00× |
| A2 | Translatotron [3] | Autoregressive | $T_{\text{mel}}$ | 79M | 16.96 | 11.25 | 2.32× |
| *Two-pass autoregressive decoding* | | | | | | | |
| B1 | UnitY [7] | Beam=(10, 1) | $T_{\text{phone}} + T_{\text{unit}}$ | 64M | 24.09 | 24.29 | 1.43× |
| B2 | Translatotron 2 [6] | Beam=10 | $T_{\text{phone}} + T_{\text{mel}}$ | 87M | **25.21** | **24.39** | 1.42× |
| *Single-pass non-autoregressive decoding* | | | | | | | |
| C1♣ | TranSpeech [9] | Iteration | 5 | 67M | 17.24 | / | 11.04× |
| C2♣ | + b=15 + NPD† | Iteration | 15 | 67M | 18.39 | / | 2.53× |
| C3♠ | TranSpeech [9] | Iteration | 5 | 67M | 16.38 | 16.49 | 12.45× |
| C4♠ | + b=15 + NPD† | Iteration | 15 | 67M | 19.05 | 18.60 | 3.35× |
| *Two-pass non-autoregressive decoding* | | | | | | | |
| D1 | **DASpeech** | Lookahead | $1 + 1$ | 93M | 24.71** | 24.45** | **18.53×** |
| D2 | ($\lambda = 0.5$) | Joint-Viterbi | $1 + 1$ | 93M | **25.03**** | **25.26**** | 16.29× |
| D3 | **DASpeech** | Lookahead | $1 + 1$ | 93M | 24.41** | 24.17** | 18.45× |
| D4 | ($\lambda = 1.0$) | Joint-Viterbi | $1 + 1$ | 93M | 24.80** | 24.48** | 15.65× |
| *Cascaded systems* | | | | | | | |
| E1 | S2T + FastSpeech 2 | Beam=10 | $T_{\text{phone}} + 1$ | 49M+41M | 24.71 | 24.49 | / |
| E2 | DAT + FastSpeech 2 | Lookahead | $1 + 1$ | 51M+41M | 22.19 | 22.10 | / |
| E3 | ($\lambda = 0.5$) | Joint-Viterbi | $1 + 1$ | 51M+41M | 22.80 | 22.75 | / |
| E4 | DAT + FastSpeech 2 | Lookahead | $1 + 1$ | 51M+41M | 22.68 | 22.57 | / |
| E5 | ($\lambda = 1.0$) | Joint-Viterbi | $1 + 1$ | 51M+41M | 23.20 | 23.15 | / |

the linguistic decoder using the speech-to-subword task, followed by pretraining on the speech-to-phoneme task. In the second stage of pretraining, the embedding and output projection matrices of the decoder are replaced and trained from scratch to accommodate changes in the vocabulary. We employ this pretraining strategy for DASpeech, UnitY and Translatotron 2 in the multilingual experiment. We learn the subword vocabulary with a size of 6K using the SentencePiece toolkit.

We also adopt the glancing strategy [25] during training, which shows effectiveness in alleviating the multi-modality problem for NAT. It first assigns target tokens to appropriate vertices following the most probable path $\hat{A} = \arg\max_{A \in \Gamma} P_\theta(Y, A|X)$, and then masks some tokens. We linearly anneal the unmasking ratio $\tau$ from 0.5 to 0.1 during pretraining and fix $\tau$ to 0.1 during finetuning.

**Evaluation** During finetuning, we save checkpoints every 2000 steps and average the last 5 checkpoints for evaluation. We use the open-source ASR-BLEU toolkit[7] to evaluate the translation quality. The translated speech is first transcribed into text using a pretrained ASR model. SacreBLEU [26] is then used to compute the BLEU score [27] and the statistical significance of translation results. The decoding speedup is measured on the `test` set using 1 RTX 3090 GPU with a batch size of 1.

**Baseline Systems** We implement the following baseline systems for comparison. More details about the model architectures and hyperparameters can be found in Appendix A.

• **S2UT** [5] Speech-to-unit translation (S2UT) model generates discrete units corresponding to the target speech with a sequence-to-sequence model. We introduce the auxiliary task of predicting target phonemes to help the model converge.

---

[7] https://github.com/facebookresearch/fairseq/tree/ust/examples/speech_to_speech/asr_bleu

- **Translatotron** [3] Translatotron generates the target mel-spectrogram with a sequence-to-sequence model. We also introduce the auxiliary task of predicting the target phonemes.

- **UnitY** [7] UnitY is a two-pass model which generates target phonemes and discrete units successively[8]. We remove the R-Drop training [28] for simplification. We pretrain the speech encoder and first-pass decoder on the S2TT task.

- **Translatotron 2** [6] Translatotron 2 is a two-pass model which generates target phonemes and mel-spectrograms successively. We enhance Translatotron 2 by replacing LSTM with Transformer, and introducing an additional encoder between two decoders following Inaguma et al. [7]. The speech encoder and first-pass decoder are pretrained on the S2TT task.

- **TranSpeech** [9] TranSpeech is the first non-autoregressive S2ST model that generates target discrete units in parallel. To alleviate the acoustic multi-modality problem, TranSpeech introduces bilateral perturbation (BiP) to disentangle the acoustic variations from the discrete units. We re-implement TranSpeech following the configurations in the original paper.

- **S2T + FastSpeech 2** The cascaded system that combines an autoregressive S2TT model and FastSpeech 2. The S2T model contains 12 Conformer layers and 4 Transformer decoder layers, which is also used in UnitY and Translatotron 2 pretraining.

- **DAT + FastSpeech 2** The cascaded system that combines the S2TT DA-Transformer model and FastSpeech 2. Both models are used in DASpeech pretraining.

## 4.2 Main Results

Table 1 summarizes the results on the CVSS-C Fr→En and CVSS-T Fr→En datasets. **(1)** Compared with previous autoregressive models, DASpeech (`D1-D4`) obviously surpasses single-pass models (`A1, A2`) and achieves comparable or even better performance than two-pass models (`B1, B2`), while preserving up to 18.53 times decoding speedup compared to S2UT. **(2)** Compared with the previous NAR model TranSpeech (`C1-C4`), DASpeech does not rely on knowledge distillation and iterative decoding, achieving significant advantages in both translation quality and decoding speedup. **(3)** DASpeech obviously outperforms the corresponding cascaded systems (`D1-D4` vs. `E2-E5`), demonstrating the effectiveness of our expect-path training approach. We also find that the cascaded model prefers larger graph size ($\lambda = 1.0$ is better) while DASpeech prefers smaller graph size ($\lambda = 0.5$ is better). We think the reason is that a larger graph size can improve S2TT performance, but it also makes end-to-end training more challenging. We further study the effects of the graph size in Appendix C. **(4)** On the CVSS-T dataset, which includes target speeches from various speakers, we observe a performance degradation in Translatotron and Translatotron 2 as the target mel-spectrogram becomes more difficult to predict. In contrast, DASpeech still performs well since its acoustic decoder explicitly incorporates variation information to alleviate the acoustic multi-modality, demonstrating the potential of DASpeech in handling complex and diverse target speeches.

Table 2 shows the results on CVSS-C dataset of the multilingual X→En S2ST model. We report the average ASR-BLEU scores on all languages, as well as the average scores on high/middle/low-resource languages[9]. We find that DASpeech still obviously outperforms S2UT but performs slightly worse than Translatotron 2 and UnitY in the multilingual setting, with an average gap of about 1.3 ASR-BLEU compared to Translatotron 2. However, DASpeech has about 13 times decoding speedup compared to Translatotron 2, achieving a better quality-speed trade-off.

Table 2: ASR-BLEU scores on CVSS-C `test` sets of the multilingual X→En S2ST model.

| Models | | Avg. | High | Mid | Low |
|---|---|---|---|---|---|
| S2UT [5] | | 5.15 | 16.74 | 6.24 | 0.84 |
| UnitY [7] | | 8.15 | 24.97 | 9.78 | 1.86 |
| Translatotron 2 [6] | | **8.74** | **25.92** | **11.07** | **2.04** |
| **DASpeech** | + Lookahead | 7.42 | 22.84 | 9.51 | 1.41 |
| ($\lambda = 0.5$) | + Joint-Viterbi | 7.43 | 22.80 | 9.49 | 1.45 |

Table 3: ASR-BLEU scores on the CVSS-C Fr→En `test` set with best-path training and expect-path training.

| Models | | Best | Expect | $\Delta$ |
|---|---|---|---|---|
| **DASpeech** | + Lookahead | 24.45 | 24.71 | +0.26 |
| ($\lambda = 0.5$) | + Joint-Viterbi | 24.84 | 25.03 | +0.19 |
| **DASpeech** | + Lookahead | 24.18 | 24.41 | +0.23 |
| ($\lambda = 1.0$) | + Joint-Viterbi | 24.46 | 24.80 | +0.34 |

---

[8]Note that the original UnitY uses subwords instead of phonemes. Here we use phonemes just for consistency with other systems.

[9]The detailed results of each language pair can be found in Table 6 in Appendix B.

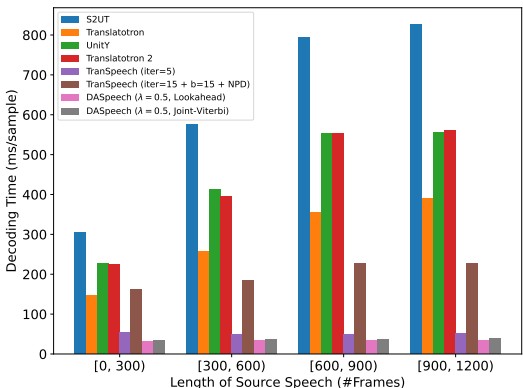
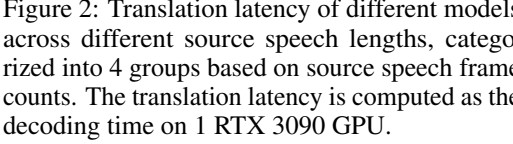
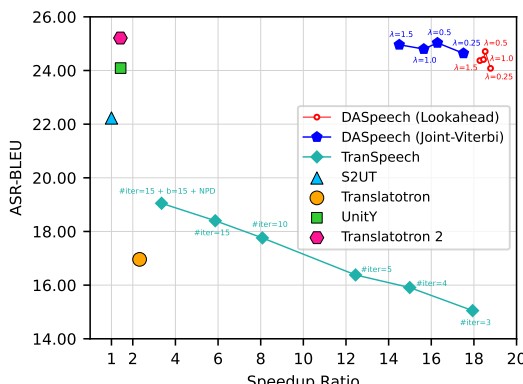

Figure 2: Translation latency of different models across different source speech lengths, categorized into 4 groups based on source speech frame counts. The translation latency is computed as the decoding time on 1 RTX 3090 GPU.

Figure 3: Trade-off between translation quality and decoding speed. X-axis represents the speedup ratio relative to S2UT, and Y-axis represents ASR-BLEU. The upper right represents better trade-off.

### 4.3 Alternative Training Approach: Best-Path Training

In addition to the expect-path training approach proposed in Section 3.2, we also experiment with a simpler approach: *Best-Path Training*. The core idea is to select the most probable path $\hat{A} = \arg\max_{A \in \Gamma} P_\theta(Y, A|X)$ via Viterbi algorithm [20], and take the hidden states on path $\hat{A} = (\hat{a}_1, ..., \hat{a}_M)$ as input to the acoustic decoder, i.e., $z_i = v_{\hat{a}_i}$. As shown in Table 3, best-path training also performs well but is inferior to expect-path training. We attribute this to the fact that when using best-path training, only hidden states on the most probable path participate in TTS training, which may result in insufficient training for the remaining hidden states. In contrast, all hidden states participate in TTS training with our expect-path training, which achieves better performance. The time complexity of Viterbi algorithm used in the best-path training is also $\mathcal{O}(ML^2)$. More details about the best-path training can be found in Appendix E.

Table 4: Average speaker similarity on CVSS-T Fr→En `test` set.

| Models | Speaker Similarity |
|---|---|
| Ground Truth | 0.48 |
| *Unit-based S2ST* | |
| S2UT [5] | 0.03 |
| UnitY [7] | 0.03 |
| TranSpeech [9] | 0.03 |
| *Mel-spectrogram-based S2ST* | |
| Translatotron [3] | 0.04 |
| Translatotron 2 [6] | 0.05 |
| **DASpeech** ($\lambda = 0.5$) + Lookahead | **0.14** |
| + Joint-Viterbi | 0.10 |

### 4.4 Analysis of Decoding Speed

In this section, we provide more detailed analysis of decoding speed. Figure 2 shows the translation latency of different models for speech inputs of different lengths. The results indicate that the translation latency of autoregressive models (S2UT, Translatotron, UnitY, and Translatotron 2) significantly increases with the length of the source speech. In contrast, the translation latency of non-autoregressive models (TranSpeech and DASpeech) are hardly affected by the source speech length. When translating longer speech inputs, DASpeech's decoding speed can reach more than 20 times that of S2UT. Furthermore, we illustrate the quality-speed trade-off of different models in Figure 3. By adjusting the hyperparameter $\lambda$, the translation quality and decoding latency will change. Specifically, as $\lambda$ increases, the decoding latency of the model will increase, and it achieves the best translation quality when $\lambda = 0.5$. It is evident that DASpeech achieves the best trade-off between translation quality and decoding latency among all models. We further study the speedup under batch decoding in Appendix D.

### 4.5 Voice Preservation

In this section, we investigate the voice preservation ability of direct S2ST models on the CVSS-T Fr→En dataset, where target speeches are in voices transferred from source speeches. Specifically, we use a pretrained speaker verification model[10] [29] to extract the speaker embedding of the source

---

[10] https://github.com/yistLin/dvector

speech and generated target speech. We define the cosine similarity between source and target speaker embeddings as *speaker similarity*, and report the average speaker similarity on the `test` set in Table 4. We find that: **(1)** unit-based S2ST model can not preserve the speaker's voice since discrete units contain little speaker information; and **(2)** DASpeech can better preserve the speaker's voice than Translatotron and Translatotron 2, since its acoustic decoder explicitly introduces variation information of the target speech, allowing the model to learn more complex target distribution.

## 5 Related Work

**Direct Speech-to-Speech Translation** Speech-to-speech translation (S2ST) extends speech-to-text translation [30–33] which further synthesizes the target speech. Translatotron [3] is the first S2ST model that directly generates target mel-spectrograms from the source speech. Since continuous speech features contain a lot of variance information that makes training challenging, Tjandra et al. [34], Zhang et al. [35] use the discrete tokens derived from a VQ-VAE model [36] as the target. Lee et al. [5, 37] extend this research line by leveraging discrete units derived from the pretrained HuBERT model [38] as the target. To further reduce the learning difficulty, Inaguma et al. [7], Jia et al. [6], Chen et al. [39] introduce a two-pass architecture that generates target text and target speech successively. To address the data scarcity issue, some techniques like pretraining and data augmentation are used to enhance S2ST [8, 40–44]. Huang et al. [9] proposes the first non-autoregressive S2ST model which achieves faster decoding speed. Our DASpeech extends this line of research and achieves better translation quality and faster decoding speed.

**Non-autoregressive Machine Translation** Machine translation based on autoregressive decoding usually has a high decoding latency [45]. Gu et al. [10] first proposes NAT for faster decoding speed. To alleviate the multi-modality problem in NAT, many approaches have been proposed [46] like knowledge distillation [47–49], latent-variable models [50, 51], learning latent alignments [52–56], sequence-level training [57, 58], and curriculum learning [25]. Recently, Huang et al. [11] introduce DA-Transformer, which models different translations with DAG to alleviate the multi-modality problem, achieving competitive results with autoregressive models. Ma et al. [59], Gui et al. [60] further enhance DA-Transformer with fuzzy alignment and probabilistic context-free grammar.

**Non-autoregressive Text-to-Speech** Ren et al. [61], Peng et al. [62] first propose non-autoregressive TTS that generates mel-spectrograms in parallel. FastSpeech 2 [12] explicitly models variance information to alleviate the issue of acoustic multi-modality. Many subsequent works enhance non-autoregressive TTS with more powerful generative models like variational auto-encoder (VAE) [63, 64], normalizing flows [65–67], and denoising diffusion probabilistic models (DDPM) [68, 69]. DASpeech adopts the design of FastSpeech 2 for training stability and good voice quality.

## 6 Conclusion

In this paper, we introduce DASpeech, a non-autoregressive two-pass direct S2ST model. DASpeech is built upon DA-Transformer and FastSpeech 2, and we propose an expect-path training approach to train the model end-to-end. DASpeech achieves comparable or even better performance than the state-of-the-art S2ST model Translatotron 2, while maintaining up to $18.53\times$ speedup compared to the autoregressive model. DASpeech also significantly outperforms previous non-autoregressive model in both translation quality and decoding speed. In the future, we will investigate how to enhance DASpeech using techniques like pretraining and data augmentation.

## 7 Limitations & Broader Impacts

**Limitations** Although DASpeech achieves impressive performance in both translation quality and decoding speed, it still has some limitations: (1) the translation quality of DASpeech still lags behind Translatotron 2 in the multilingual setting; (2) the training cost of DASpeech is higher than Translatotron 2 (96 vs. 18 GPU hours) since it requires dynamic programming during training; and (3) the outputs of DASpeech are not always reliable, especially for some low-resource languages.

**Broader Impacts** In our experiments, we find that DASpeech emerges with the ability to maintain the speaker identity during translation. It raises potential risks in terms of model misuse, such as mimicking a particular speaker or voice identification spoofing.

## Acknowledgements

We thank all the anonymous reviewers for their insightful and valuable comments.

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

# A Details of Baseline Models

In our experiments, we implement five baseline systems using *fairseq*: S2UT, Translatotron, UnitY, Translatotron 2, and TranSpeech. We reproduce TranSpeech with their open-source implementations[11]. In this section, we mainly introduce the configurations of the other four baseline systems.

Figure 4 shows the model architectures of these models. In terms of model architecture, S2UT and Translatotron are single-pass S2ST models while UnitY and Translatotron 2 are two-pass S2ST models. In terms of predicted targets, S2UT and UnitY predict discrete units while Translatotron and Translatotron 2 predict mel-spectrograms. Below we describe the details of each model. The detailed hyperparameters can be found in Table 5.

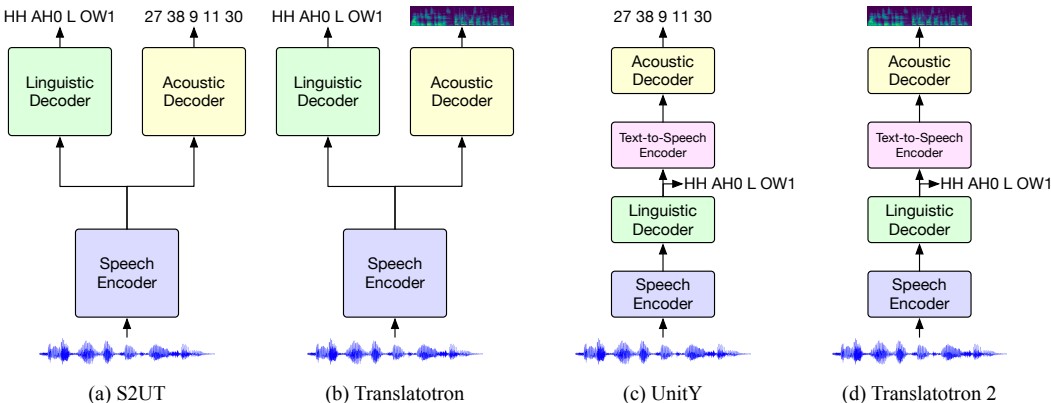

Figure 4: Overview of baseline models.

**S2UT** Our implemented S2UT model includes three parts: a speech encoder, a linguistic decoder, and an acoustic decoder. The speech encoder is the same as DASpeech. The linguistic decoder is appended to the top layer of the speech encoder for multi-task learning, which predicts the target phonemes during training. The acoustic decoder generates the reduced discrete units derived from the 11-th layer of the pretrained mHuBERT model[12]. We do not include other auxiliary tasks and remove CTC decoding in Lee et al. [5] for simplification. The model is trained from scratch for 100k steps. We use beam search with a beam size of 10.

**Translatotron** The speech encoder and linguistic decoder of Translatotron are the same as S2UT. The acoustic decoder generates mel-spectrograms autoregressively. The pre-net dimension is 32 and the reduction factor of the acoustic decoder is 5. The model is trained from scratch for 100k steps.

**UnitY** UnitY is a two-pass model that includes four parts: a speech encoder, a linguistic decoder, a text-to-speech encoder, and an acoustic decoder. The architecture of the speech encoder, linguistic decoder, and acoustic decoder are the same as S2UT. The additional text-to-speech encoder is used to bridge the gap in representations between two decoders. We remove R-Drop training for simplification. We first conduct S2TT pretraining and finetune the model for 50k steps. We set the beam size of the first-pass and second-pass decoder to 10 and 1, respectively.

**Translatotron 2** The model architecture of Translatotron 2 is similar to UnitY except that the second decoder generates mel-spectrograms rather than discrete units. The reduction factor of the acoustic decoder is set to 5. We first conduct S2TT pretraining and finetune the model for 50k steps. The beam size is set to 10 for the first-pass decoder.

For all the above models, we save checkpoints every 2000 steps and average the last 5 checkpoints for evaluation, which is the same as DASpeech. For S2UT and UnitY, we use the pretrained unit-based HiFi-GAN[13] vocoder to synthesize waveform. For Translatotron and Translatotron 2, we use the same pretrained HiFi-GAN vocoder as DASpeech.

---

[11]https://github.com/Rongjiehuang/TranSpeech
[12]https://dl.fbaipublicfiles.com/hubert/mhubert_base_vp_en_es_fr_it3_L11_km1000.bin
[13]https://dl.fbaipublicfiles.com/fairseq/speech_to_speech/vocoder/code_hifigan/mhubert_vp_en_es_fr_it3_400k_layer11_km1000_lj/g_00500000

Table 5: Hyperparameters of DASpeech and baseline models.

| Hyperparameters | | S2UT | Translatotron | UnitY | Translatotron 2 | DASpeech |
|---|---|---|---|---|---|---|
| Speech Encoder | conv_kernel_sizes | (5, 5) | (5, 5) | (5, 5) | (5, 5) | (5, 5) |
| | encoder_type | conformer | conformer | conformer | conformer | conformer |
| | encoder_layers | 12 | 12 | 12 | 12 | 12 |
| | encoder_embed_dim | 256 | 256 | 256 | 256 | 256 |
| | encoder_ffn_embed_dim | 2048 | 2048 | 2048 | 2048 | 2048 |
| | encoder_attention_heads | 4 | 4 | 4 | 4 | 4 |
| | encoder_pos_enc_type | relative | relative | relative | relative | relative |
| | depthwise_conv_kernel_size | 31 | 31 | 31 | 31 | 31 |
| Linguistic Decoder | decoder_layers | 4 | 4 | 4 | 4 | 4 |
| | decoder_embed_dim | 512 | 512 | 512 | 512 | 512 |
| | decoder_ffn_embed_dim | 2048 | 2048 | 2048 | 2048 | 2048 |
| | decoder_attention_heads | 8 | 8 | 8 | 8 | 8 |
| | label_smoothing | 0.1 | 0.1 | 0.1 | 0.1 | 0.0 |
| | s2t_loss_weight | 8.0 | 0.1 | 8.0 | 0.1 | 1.0 |
| Text-to-Speech Encoder | encoder_layers | - | - | 2 | 2 | - |
| | encoder_embed_dim | - | - | 512 | 512 | - |
| | encoder_ffn_embed_dim | - | - | 2048 | 2048 | - |
| | encoder_attention_heads | - | - | 8 | 8 | - |
| Acoustic Decoder | decoder_layers | 6 | 6 | 2 | 6 | 8 |
| | decoder_embed_dim | 512 | 512 | 512 | 512 | 256 |
| | decoder_ffn_embed_dim | 2048 | 2048 | 2048 | 2048 | 1024 |
| | decoder_attention_heads | 8 | 8 | 8 | 8 | 4 |
| | label_smoothing | 0.1 | - | 0.1 | - | - |
| | n_frames_per_step | 1 | 5 | 1 | 5 | 1 |
| | unit_dictionary_size | 1000 | - | 1000 | - | - |
| | var_pred_hidden_dim | - | - | - | - | 256 |
| | var_pred_kernel_size | - | - | - | - | 3 |
| | var_pred_dropout | - | - | - | - | 0.5 |
| | s2s_loss_weight | 1.0 | 1.0 | 1.0 | 1.0 | 5.0 |
| Training | lr | 1e-3 | 1e-3 | 1e-3 | 1e-3 | 1e-3 |
| | lr_scheduler | inverse_sqrt | inverse_sqrt | inverse_sqrt | inverse_sqrt | inverse_sqrt |
| | warmup_updates | 4000 | 4000 | 4000 | 4000 | 4000 |
| | warmup_init_lr | 1e-7 | 1e-7 | 1e-7 | 1e-7 | 1e-7 |
| | optimizer | Adam | Adam | Adam | Adam | Adam |
| | dropout | 0.1 | 0.1 | 0.1 | 0.1 | 0.1 |
| | max_tokens | 40k×4 | 40k×4 | 40k×4 | 40k×4 | 40k×8 |
| | weight_decay | 0.0 | 0.0 | 0.0 | 0.0 | 0.01 |
| | clip_norm | 1.0 | 1.0 | 1.0 | 1.0 | 1.0 |
| | max_update | 100k | 100k | 50k | 50k | 50k |

# B  Detailed Results on CVSS-C X→En Datasets

Table 6 summarizes the detailed results of each language pair on CVSS-C `test` sets of the multilingual X→En S2ST models.

Table 6: Results on CVSS-C `test` sets of the multilingual X→En S2ST models.

| Models | | Avg. | High | | | | Mid | | | | |
|---|---|---|---|---|---|---|---|---|---|---|---|
| | | | Fr | De | Ca | Es | Fa | It | Ru | Zh | Pt |
| S2UT [5] | | 5.15 | 19.65 | 13.35 | 15.37 | 18.58 | 1.43 | 14.47 | 7.94 | 0.93 | 6.42 |
| UnitY [7] | | 8.15 | 27.27 | 20.81 | 24.22 | 27.58 | 3.63 | 21.68 | 10.86 | 4.16 | 8.56 |
| Translatotron 2 [6] | | 8.74 | 28.04 | 21.54 | 25.34 | 28.77 | 4.23 | 23.66 | 13.41 | 4.49 | 9.54 |
| **DASpeech** | + Lookahead | 7.42 | 25.43 | 17.87 | 22.58 | 25.49 | 3.01 | 20.80 | 12.96 | 2.86 | 7.90 |
| ($\lambda = 0.5$) | + Joint-Viterbi | 7.43 | 25.39 | 18.36 | 22.33 | 25.10 | 2.81 | 20.76 | 12.94 | 3.05 | 7.89 |

| Models | | Low | | | | | | | | | | | |
|---|---|---|---|---|---|---|---|---|---|---|---|---|---|
| | | Nl | Tr | Et | Mn | Ar | Lv | Sl | Sv | Cy | Ta | Ja | Id |
| S2UT [5] | | 4.67 | 0.52 | 0.36 | 0.14 | 0.56 | 0.39 | 0.73 | 1.28 | 0.66 | 0.17 | 0.20 | 0.38 |
| UnitY [7] | | 10.60 | 3.79 | 1.07 | 0.12 | 0.78 | 1.50 | 0.81 | 1.38 | 1.74 | 0.10 | 0.15 | 0.27 |
| Translatotron 2 [6] | | 11.17 | 4.58 | 1.12 | 0.32 | 1.35 | 1.37 | 0.93 | 1.49 | 1.50 | 0.10 | 0.22 | 0.33 |
| **DASpeech** | + Lookahead | 9.04 | 1.75 | 0.04 | 0.08 | 0.64 | 1.43 | 1.20 | 1.33 | 0.70 | 0.09 | 0.29 | 0.29 |
| ($\lambda = 0.5$) | + Joint-Viterbi | 9.43 | 1.66 | 0.07 | 0.08 | 0.48 | 1.48 | 1.30 | 1.30 | 0.85 | 0.09 | 0.31 | 0.32 |

## C   Effects of the Graph Size

In this section, we investigate how the graph size affects the performance. We vary the size factor $\lambda$ from 0.25 to 1.5, and measure the translation quality of both the S2TT DA-Transformer model and DASpeech on the CVSS-C Fr→En `test` set. As shown in Figures 5 and 6, we observe that the performance of S2TT DA-Transformer keeps increasing as the graph size gets larger, which is consistent with the observations in machine translation [11, 59]. However, DASpeech performs best at $\lambda = 0.5$ and shows a performance drop at larger $\lambda$. We speculate that this is because larger graph size makes end-to-end training more challenging. We will investigate this issue in the future.

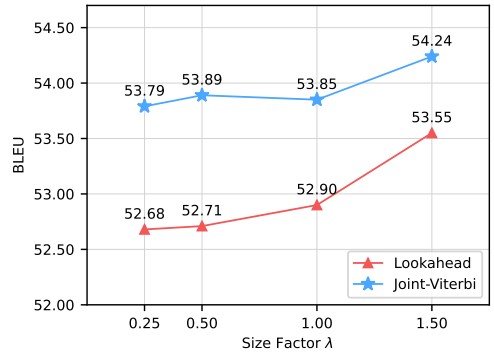
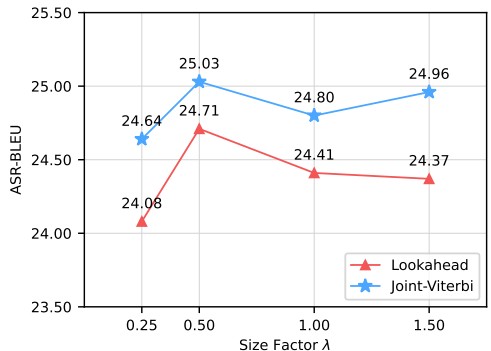

Figure 5: Phoneme-level BLEU scores of the S2TT DA-Transformer under different size factor $\lambda$.

Figure 6: ASR-BLEU scores of DASpeech under different size factor $\lambda$.

## D   Speedup Under Batch Decoding

As Gu and Kong [46] pointed out, the speed benefits of non-autoregressive models may degrade during batch decoding. To better understand this problem, we evaluate the speedup ratio under different decoding batch sizes. As shown in Figure 7, the speedup ratio keeps dropping as the decoding batch size increases. Nevertheless, DASpeech ($\lambda = 0.5$ with Joint-Viterbi decoding) still achieves more than $6\times$ speedup with a decoding batch size of 64 and maintains comparable performance with Translatotron 2.

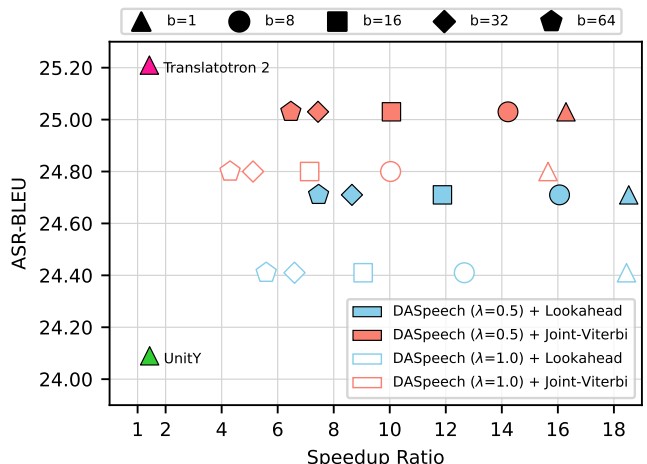

Figure 7: Speedup ratio compared to S2UT baseline (not shown in the figure) and ASR-BLEU score on the CVSS-C Fr→En `test` set with different batch decoding sizes ($b \in \{1, 8, 16, 32, 64\}$).

# E   Best-Path Training

Best-path-training selects the most probable path $\hat{A} = (\hat{a}_1, ..., \hat{a}_M)$ and takes the hidden states on path $\hat{A}$ as input to the acoustic decoder. Formally, given the target phoneme sequence $Y$, we can find the most probable path $\hat{A} = \arg\max_{A \in \Gamma} P_\theta(Y, A|X)$ via Viterbi algorithm [20]. Specifically, we use $\delta_i(j)$ to denote the probability of the most probable path so far $(\hat{a}_1, ..., \hat{a}_i)$ with $\hat{a}_i = j$ that generates $(y_1, ..., y_i)$. Considering the definition of $a_1 = 1$, we have $\delta_1(1) = \mathbf{P}_{1,y_1}$ and $\delta_1(1 < j \leq L) = 0$. For $i > 1$, we can sequentially calculate $\delta_i(\cdot)$ from its previous step $\delta_{i-1}(\cdot)$ due to the Markov property:

$$\delta_i(j) = \max_{k<j}(\delta_{i-1}(k) \cdot \mathbf{E}_{k,j} \cdot \mathbf{P}_{j,y_i}), \tag{16}$$

$$\phi_i(j) = \arg\max_{k<j}(\delta_{i-1}(k) \cdot \mathbf{E}_{k,j} \cdot \mathbf{P}_{j,y_i}), \tag{17}$$

where $\phi_i(j)$ stores $\hat{a}_{i-1}$ of the most probable path so far $(\hat{a}_1, ..., \hat{a}_{i-1}, \hat{a}_i = j)$. After $M$ iterations, we can obtain the most probable path by backtracking from $\hat{a}_M = L$:

$$\hat{a}_i = \phi_{i+1}(\hat{a}_{i+1}). \tag{18}$$

Finally, we select the hidden states on the most probable path, i.e., $\mathbf{z}_i = \mathbf{v}_{\hat{a}_i}$, as the input sequence of the acoustic decoder.

