# OpenReview forum: "DASpeech: Directed Acyclic Transformer for Fast and High-quality Speech-to-Speech Translation"
_NeurIPS.cc/2023/Conference — NeurIPS 2023 poster_

### Official Review · Reviewer_WJ3s · 2023-07-03

**Soundness:** 4 excellent
**Presentation:** 4 excellent
**Contribution:** 4 excellent
**Rating:** 8
**Confidence:** 5

**Summary:**

This paper presents a non-autoregressive direct speech-to-speech translation model based on Directed Acyclic Transformer. Experiments are conducted on CVSS-C and CVSS-T datasets, demonstrating translation performance comparable to the SOTA (Translatotron 2) but more than 16x speed up. It also demonstrates capacity of voice preservation during translation.

**Strengths:**

1) The proposed approach is sound
2) Although the proposed approach is largely a combination of existing works (Directed Acyclic Transformer for NMT and Translatotron 2 for S2ST), its effective adoption in S2ST is novel.
3) The experimental results on translation quality, speed, and voice preservation are all promising.
4) Audio samples accompanying the paper sound consistent with the quantitive evaluation conducted in the paper.

**Weaknesses:**

1) The main results (Table 1) is conducted on English-French translation, which is an easy case as the order of the two languages are very close. It would be helpful to include more results on languages with significant difference on ordering, such as English-German. Compared to autoregressive model, one potential disadvantage of non-autogressive model is it may not handle reordering well.

**Questions:**

1) Voice Preservation evaluation -- from the audio samples, it can been heard clearly as the unit-based approaches don't preserve voice, but DASpeech and Translatotron 1 & 2 can (as stated by the paper). However, looking at the numbers in Table 4, the similarities from Translatotron 1 & 2 are very close to unit-based approach (with DASpeech signicantly higher) -- any explaination?

**Limitations:**

Yes.

---

> ### Author Rebuttal · Authors · 2023-08-09
>
> We are deeply grateful for your positive evaluation of our work. Your valuable comments have been carefully considered, and we have provided responses to each of them below.
>
> 1. **Re: It would be helpful to include more results on languages with significant differences in ordering, such as English-German.**
>
> Thank you for your suggestions. Based on your advice, we conducted additional experiments on the CVSS-C De-En dataset. **Please refer to the "Global Response" for details.** The experiments confirmed that DASpeech has also outperformed the state-of-the-art performance in the De-En direction, confirming the effectiveness of DASpeech.
>
> 2. **Re: Explaination about the voice preservation evaluation**
>
> Thank you for bringing up this issue. We have also observed that Translatotron 1/2 exhibit only a slight improvement in speaker similarity compared to unit-based models. From the audio samples, it is evident that while Translatotron1/2 possess some voice cloning capability, the generated audio quality is not very high and often contains significant background noise. Since we extracted speaker embeddings using a speaker verification model trained on real audio, **we suspect that the presence of noise in the generated audio may negatively impact the calculation of speaker similarity due to distribution shift between training and decoding of the speaker verification model.** In contrast, DASpeech generates audio that is more natural, fluent, and with reduced noise. Additionally, it exhibits stronger voice cloning capabilities. As a result, it significantly outperforms other models in terms of speaker similarity.

---

> > ### Comment · Reviewer_WJ3s · 2023-08-17
> >
> > I would like to thank the authors for the responses and the follow up experiments. The new results on De-En make the paper significantly more convincing. Therefore, I bounced my rating from 7 to 8.

---

> > > ### Author Response · Authors · 2023-08-20
> > > **Thank you for raising the score!**
> > >
> > > We greatly appreciate your consideration in raising the score. Thank you very much!

---

### Official Review · Reviewer_PNQE · 2023-07-06

**Soundness:** 3 good
**Presentation:** 3 good
**Contribution:** 3 good
**Rating:** 5
**Confidence:** 5

**Summary:**

DASpeech is a non-autoregressive direct speech-to-speech translation (S2ST) model. The model adopts a two-pass architecture, where a linguistic decoder first generates the target text, and an acoustic decoder then generates the target speech based on the hidden states of the linguistic decoder. The linguistic decoder is based on the DA-Transformer model, which models translations with a directed acyclic graph (DAG). The acoustic decoder is based on the FastSpeech 2 model.

Experiments on the CVSS benchmark demonstrate that DASpeech can achieve comparable or even better performance than the state-of-the-art S2ST model Translatotron 2, while preserving up to 18.53x speedup compared to the autoregressive baseline model. Compared to the previous non-autoregressive S2ST model, DASpeech does not rely on knowledge distillation and iterative decoding, achieving significant improvements in both translation quality and decoding speed. Furthermore, DASpeech shows the ability to preserve the speaker's voice of the source speech during translation.

**Strengths:**

1. A new direct Speech to Speech translation model that use Conformer Encoder, Non-autoregressive Linguistic Decoder, Directed
Acyclic Graph and Non-autoregressive Acoustic Decoder.
2. Efficient architecture for Speech to Speech translation - up to 18.53 speedup compared to the autoregressive baseline model.
3. Get comparable results to state of the art models in terms of translation quality (BLEU)

A new direct Speech to Speech translation model:
This model is a neural network that can translate speech from one language to another directly, without the need for an intermediate step of transcribing the speech to text. This makes it more efficient and faster than traditional speech translation systems, which require two separate models: one for speech recognition and one for machine translation. The model uses a Conformer encoder, which is a type of neural network that has been shown to be effective for speech recognition tasks. The encoder converts the input speech signal into a sequence of hidden representations. These representations are then decoded by a non-autoregressive linguistic decoder, which generates the translation sentence one word at a time. The model also uses a directed acyclic graph (DAG) and a non-autoregressive acoustic decoder. The DAG is a data structure that allows the model to predict multiple words at a time, which can further improve the efficiency of the translation process. The non-autoregressive acoustic decoder generates the output speech signal one frame at a time, which helps to preserve the naturalness of the audio.

Efficient architecture for Speech to Speech translation:
The model's architecture is designed to be efficient, both in terms of memory and speed. The Conformer encoder is a relatively lightweight model, which helps to reduce the memory requirements of the system. The non-autoregressive linguistic decoder and acoustic decoder are also designed to be efficient, which helps to speed up the translation process. The model has been shown to be up to 18.53 times faster than the autoregressive baseline model. This makes it a promising candidate for real-time speech translation applications.

Comparable results to state of the art models in terms of translation quality (BLEU):
The model has also been shown to achieve comparable results to state-of-the-art models in terms of translation quality (BLEU).


**Weaknesses:**

.

**Questions:**

1. The proposed method only evaluate for Fr-En languages. Can you evaluate the proposed method with more languages?
2. What is this model offer other then speed-up? The results is comparable to T2. Can the translation results improve?
3. Can you add ablation analysis of each component?

**Limitations:**

.

---

> ### Author Rebuttal · Authors · 2023-08-09
>
> Thank you for providing your valuable and constructive feedback! Below, you will find our responses to each comment.
>
> 1. **Re: Can you evaluate the proposed method with more languages?**
>
> Thank you for your suggestions. Based on your advice, we conducted additional experiments on the CVSS-C De-En dataset. **Please refer to the "Global Response" for details.** The experiments confirmed that DASpeech has also outperformed the state-of-the-art performance in the De-En direction, confirming the effectiveness of DASpeech.
>
> 2. **Re: What is this model offer other then speed-up? The results is comparable to T2. Can the translation results improve?**
>
> From the current results, DASpeech offers not only a significant speed advantage over Translatotron 2 but also demonstrates superior modeling capabilities for complex target voices. In Table 1's experimental results, when the target voice contains only a single speaker (CVSS-C dataset), DASpeech's translation quality slightly falls behind Translatotron 2 (25.03 vs. 25.21). However, when the target voice includes multiple speakers (CVSS-T dataset), DASpeech exhibits a 0.87 ASR-BLEU improvement over Translatotron 2. This improvement is attributed to DASpeech's acoustic decoder's ability to better model the multi-modal characteristics of the speech. Audio samples provided in our paper also reveal that DASpeech produces better audio quality than Translatotron 2 on the CVSS-T dataset.
>
> Furthermore, **we believe DASpeech has even greater potential for future development.** On one hand, recent research [1] indicates that further fine-tuning the DA-Transformer with a fuzzy alignment objective can enhance translation quality, which can be beneficial for DASpeech as well. On the other hand, the training of DASpeech does not rely on a specific TTS structure, allowing us to use more powerful TTS models than FastSpeech 2, enabling us to model more diverse and complex voices. We intend to explore these directions in our future research.
>
> [1] Ma et al., 2023. Fuzzy Alignments in Directed Acyclic Graph for Non-Autoregressive Machine Translation. ICLR 2023.
>
> 3. **Re: Can you add ablation analysis of each component?**
>
> Thank you for your suggestions. We added the following two ablation experiments:
>
>   - Removing the glancing strategy during the finetuning stage.
>   - Removing TTS pretraining.
>
> The results on the CVSS-C Fr-En test set are shown in the table below. All results are based on the Lookahead decoding strategy. We observed that these two factors have a minor impact on the results. However, they still contribute to a slight improvement.
>
> | Models                              | ASR-BLEU |
> | ----------------------------------- | -------- |
> | DASpeech ($\lambda=0.5$, Lookahead) | 24.71    |
> | w/o glancing                        | 24.68    |
> | w/o TTS pretraining                 | 24.63    |
>
> In addition to this, the training of DASpeech involves two crucial hyperparameters: **the weight of the TTS loss, denoted as $\mu$, and the target-side input length coefficient for the linguistic decoder, denoted as $\lambda$.** The impact of these two hyperparameters on the CVSS-C Fr-En dataset is illustrated in the table below. All results are based on the Lookahead decoding strategy.
>
> | $\mu$                              | ASR-BLEU |
> | ----------------------------------- | -------- |
> | 1  | 23.80    |
> | 2  | 24.12    |
> | 5  | **24.71**    |
> | 10  |  collapse |
>
> | $\lambda$                     | ASR-BLEU |
> | ----------------------------------- | -------- |
> | 0.25  | 24.08    |
> | 0.5  | **24.71**    |
> | 1.0  | 24.41    |
> | 1.5 |  24.37 |
>
> Regarding the expect-path training method we proposed, we have also conducted a comparison with another strategy, best-path training, as detailed in Section 4.3 of the paper. **Apart from this, if you would like to see results for other configurations, please let us know. We would be more than happy to continue supplementing experiments during the reviewer-author discussion phase.**

---

### Official Review · Reviewer_RzGP · 2023-07-07

**Soundness:** 2 fair
**Presentation:** 3 good
**Contribution:** 2 fair
**Rating:** 5
**Confidence:** 4

**Summary:**

The paper describes a non-autoregressive direct Speech-to-Speech Translation (S2ST) model called DASpeech that achieves both high-quality translations and fast decoding speeds. DASpeech uses a two-pass architecture where a linguistic decoder generates the target text, and an acoustic decoder generates the target speech based on the hidden states of the linguistic decoder. The linguistic decoder used is the decoder of DA-Transformer, and the acoustic decoder used is FastSpeech 2. DASpeech achieves much better performance than non-autoregressive baseline and 18.53x speedup compared to autoregressive baseline models and preserves the speaker's voice of the source speech during translation.

**Strengths:**

The paper proposes a non-autoregressive direct S2ST model that achieves both high-quality translations and fast decoding speeds. The experiments show that DASpeech performs better than the baseline NAT model and is comparable to the state-of-the-art AT model in terms of translation quality.

**Weaknesses:**

However, there are some issues in the paper that need to be addressed.

Firstly, the novelty of the paper is not significant. The paper is based on the DA-Transformer model heavily borrowed from the reference [11] (i.e., NAT in text-to-text translation), and there is no breakthrough in terms of technology. Additionally, it is not clear that the proposed model can solve some unique problems in speech-to-speech translation scenario.

Secondly, some of the contributions mentioned in the paper are not well-supported by the experiments. For example, the paper claims that DASpeech can achieve comparable or better performance than SOTA, but the experimental results show that it performs worse.

Thirdly, the efficiency comparison is not in-depth enough. Although efficiency is one of the major contributions of the paper, there is a lack of detailed analysis and tables to describe the efficiency. Moreover, since the paper is a NAT model, it is not fair to compare its speed with the AT model directly. It would be more reasonable to compare the efficiency with the NAT baseline.

**Questions:**

n/a

**Limitations:**

See weakness

---

> ### Author Rebuttal · Authors · 2023-08-09
>
> Thank you for your helpful feedback! Please find below the responses to each comment.
>
> 1. **Re: The novelty of the paper is not significant. It is not clear that the proposed model can solve some unique problems in speech-to-speech translation scenario.**
>
> Thank you for pointing that out. In our opinion, applying the DA-Transformer to **speech-to-text translation** is straightforward as the prediction target remains the target text, and only the text encoder needs to be replaced with a speech encoder. **However, its application to speech-to-speech translation (S2ST) is non-trivial.** For the two-pass S2ST model, predicting the target text is just the first step; it also requires an acoustic decoder to further synthesize the target speech based on the output hidden states from the linguistic decoder. During the training process, **we need to find the decoder's hidden state sequence corresponding to the ground truth target text.** For auto-regressive models like UnitY and Translatotron 2, teacher forcing training naturally addresses this problem. However, for the DA-Transformer, which utilizes a Directed Acyclic Graph (DAG) to model multiple possible translations simultaneously, the decoder's output consists of representations for all nodes in the DAG. To find the hidden states corresponding to the ground truth text, we proposed the **"expect-path training"** method (Section 3.2). This method considers all possible paths in the DAG and calculates the posterior probabilities $P_\theta(a_i|X,Y)$ through dynamic programming. Based on these posterior probabilities, we are able to compute the **expected hidden states** corresponding to each word in the ground truth text. This enables us to jointly train the DA-Transformer and the Text-to-Speech (TTS) model for the S2ST task.
>
> **As Reviewers nvMe and WJ3s mentioned, while the novelty of our paper builds upon prior works, effectively integrating them to address a new and challenging task is also considered novel. Note that our approach is not limited to a specific TTS model.** On the contrary, our focus is on efficiently integrating the S2TT DA-Transformer and the TTS model, enabling end-to-end training for the S2ST task. As a result, theoretically, our method can also combine the S2TT DA-Transformer with other TTS models that are more powerful than FastSpeech 2, allowing for the modeling of more diverse and complex target speech. We leave this for future research. We hope that our work can serve as a strong baseline for non-autoregressive speech-to-speech translation in the future.
>
> 2. **Re: The paper claims that DASpeech can achieve comparable or better performance than SOTA, but the experimental results show that it performs worse.**
>
> We are sorry for the lack of precision in our claim. In fact, we have indeed achieved comparable or even better results than the state-of-the-art (SOTA) on the CVSS Fr-En dataset, as shown in Table 1. However, DASpeech performs slightly worse than SOTA in the multilingual scenario. Nevertheless, our results also significantly outperform the baseline model S2UT. We will modify our statement in the final version to ensure strict consistency with the experimental results.
>
> 3. **Re: Detailed analysis and tables to describe the efficiency**
>
> Thank you for your advice. Following your suggestion, we have included several additional experiments concerning decoding efficiency of DASpeech. These experiments provide further insights into the trade-off between translation quality and decoding speed of DASpeech. **You can refer to the "Global Response" for more details.**
>
> 4. **Re: It is not fair to compare its speed with the AT model directly. It would be more reasonable to compare the efficiency with the NAT baseline.**
>
> Thank you for your suggestion. Firstly, we acknowledge that NAT models have an inherent advantage in decoding speed compared to AT models. However, the parallel decoding may lead to a decline in translation quality to some extent. Therefore, we believe that comparing NAT to AT models in both translation quality and decoding speed allows us to highlight the trade-offs between different models. In the field of text-to-text translation, previous studies have also compared NAT models with autoregressive Transformers to assess decoding speed. Thus, **we consider our comparison of DASpeech with the autoregressive model S2UT to be reasonable.**
>
> Furthermore, concerning the comparison with NAT models, DASpeech also exhibits significantly improved decoding efficiency compared to TranSpeech. For instance, taking DASpeech ($\lambda=0.5$, lookahead decoding) as an example, on the CVSS-C Fr-En dataset, it achieves a decoding speed improvement of 1.49x compared to TranSpeech with 5-iterations decoding, while simultaneously improving translation quality by 8.33 ASR-BLEU. **Compared to TranSpeech with 15-iterations decoding, DASpeech achieves a decoding speed improvement of 5.53x and enhances translation quality by 5.66 ASR-BLEU.** We believe these comparative evaluations contribute to a comprehensive understanding of the strengths and limitations of different models, and they highlight the benefits of our proposed DASpeech.

---

> ### Comment · Reviewer_RzGP · 2023-08-17
> **Thanks for your additional experiments.**
>
> In the paper, there is only a single NAT baseline. Would you please compare more SOTA NAT systems in recent two years in terms of both quality and efficiency?  I may raise the score if i see more positive results in the new experiments.

---

> > ### Author Response · Authors · 2023-08-20
> > **New results from improved versions of TranSpeech with advanced NAT models.**
> >
> > Thank you for your valuable suggestions! To the best of our knowledge, TranSpeech is currently the only released non-autoregressive S2ST model. Thus, our paper only contains this NAT baseline. However, as TranSpeech is a model based on CMLM [1], it lags behind state-of-the-art NAT models in the trade-off between translation quality and decoding speed. **Following your suggestions, we have implemented several improved versions of TranSpeech with CTC-based NAT models [2].** The CTC-based NAT model captures latent alignments between model outputs and target tokens, achieving strong performance with only single forward of decoder. Notably, we only replaced the NAT model from CMLM to CTC, while preserving TranSpeech's Bilateral Perturbation (BiP) and Sequence-level Distillation (Seq-KD), which can reduce the multimodality of training data.
> >
> > **For further performance enhancement, we have incorporated the following two training techniques into the CTC-based NAT model:**
> >
> > 1. **GLAT [3]: Glancing Targets (GLAT)** is a curriculum learning strategy that allows the decoder to glance some ground truth tokens during training. The glancing ratio is dynamically adjusted based on translation error rate and training time steps.
> > 2. **NMLA [4]: Non-Monotonic Latent Alignments (NMLA)** addresses the issue of CTC modeling only monotonic alignments by introducing a loss based on n-gram matching. In our experiments, we used a 2-gram matching loss and fine-tuned models trained with CTC for 4k steps.
> >
> > Through the integration of these two training techniques, the CTC-based NAT model achieves further improvements in translation quality. For all models, we utilized argmax decoding. Experimental results on the CVSS-C Fr-En test set are shown in the table below:
> >
> > | ID    | Models                                  | \#Iter | ASR-BLEU  | Speedup    |
> > | ----- | --------------------------------------- | ------ | --------- | ---------- |
> > | 1     | S2UT                                    | T      | 22.23     | 1.00x      |
> > | 2     | TranSpeech (CMLM)                       | 5      | 16.38     | 12.45x     |
> > | 3     | TranSpeech (CMLM) + b=15 + NPD          | 15     | 19.05     | 3.35x      |
> > | **4** | ***TranSpeech (CTC)**                   | 1      | 16.85     | **18.71x** |
> > | **5** | ***TranSpeech (CTC + GLAT)**            | 1      | 18.57     | **18.71x** |
> > | **6** | ***TranSpeech (CTC + NMLA)**            | 1      | 19.30     | **18.71x** |
> > | **7** | ***TranSpeech (CTC + GLAT + NMLA)**     | 1      | 20.51     | **18.71x** |
> > | 8     | DASpeech ($\lambda=0.5$), Lookahead     | 1+1    | 24.71     | 18.53x     |
> > | 9     | DASpeech ($\lambda=0.5$), Joint-Viterbi | 1+1    | **25.03** | 16.29x     |
> >
> > *: Our implemented improved versions of TranSpeech with advanced NAT models.
> >
> > **From the table, it is evident that our improved versions of TranSpeech (4-7) exhibit substantial improvements in both translation quality and decoding speed compared to the original CMLM-based TranSpeech (2-3).** Specifically, Vanilla CTC surpasses the translation quality of CMLM's 5-iteration decoding (2 vs. 4), while CTC + GLAT approaches the translation quality of CMLM's 15-iteration decoding (3 vs. 5). The addition of NMLA training further results in a notable improvement in translation quality. Specifically, CTC + NMLA and CTC + GLAT + NMLA attain ASR-BLEU scores of 19.30 and 20.51, respectively.
> >
> > As demonstrated, when utilizing the most advanced NAT techniques, TranSpeech approaches the translation quality of the autoregressive S2UT model while maintaining an 18.71x decoding speedup. **However, even the strongest system, CTC + GLAT + NMLA, when compared to our DASpeech (7 vs. 8-9), though having a marginal advantage in terms of speedup, still lags behind by more than 4 ASR-BLEU points in translation quality.** This is attributed to DASpeech's two-pass structure, which alleviates the difficulty of predicting target speech through task decomposition, thereby leading to stronger performance. Furthermore, owing to DASpeech's inherent ability to model multimodal data distributions, there's no need to mitigate data multimodality through techniques like BiP and Seq-KD before training.
> >
> > **In conclusion, we firmly believe that DASpeech demonstrates a superior quality-speed trade-off compared to both state-of-the-art AT and NAT models.** We appreciate your feedback, which has contributed to refining our comparisons. We would greatly appreciate it if you could consider raising the score.
> >
> > **References:**
> >
> > [1] Mask-Predict: Parallel Decoding of Conditional Masked Language Models (Ghazvininejad et al., EMNLP-IJCNLP 2019)
> >
> > [2] End-to-End Non-Autoregressive Neural Machine Translation with Connectionist Temporal Classification (Libovický & Helcl, EMNLP 2018)
> >
> > [3] Glancing Transformer for Non-Autoregressive Neural Machine Translation (Qian et al., ACL-IJCNLP 2021)
> >
> > [4] Non-Monotonic Latent Alignments for CTC-Based Non-Autoregressive Machine Translation (Shao & Feng, NeurIPS 2022)

---

> > > ### Comment · Reviewer_RzGP · 2023-08-21
> > > **Thanks for the additional experiments**
> > >
> > > I am happy to raise my score.

---

> > > > ### Author Response · Authors · 2023-08-21
> > > > **Thank you for raising the score!**
> > > >
> > > > Thank you for raising the score! Once again, thank you for your timely and valuable feedback during the discussion phase.

---

### Official Review · Reviewer_nvMe · 2023-07-07

**Soundness:** 4 excellent
**Presentation:** 3 good
**Contribution:** 3 good
**Rating:** 7
**Confidence:** 4

**Summary:**

This paper proposes a method for fast and high quality speech to speech translation.
The paper adopts the two-pass decoding framework.
The first pass decoder adapts prior work on non autoregressive decoding for machine translation.
The second pass decoder reuses fastspeech 2 except that the input are the hidden states obtained by the first pass decoder.
The first pass and second pass components are pretrained on the speech to text translation task and the synthesis task respectively.
Results show competitive performance compared to the state of the art, with substantial inference speedups.
The model is also shown to be able to preserve voice when trained on a corpus with the same voice on the source and target sides.

**Strengths:**

* the approach is effective: competitive wrt SoTA with much faster inference speed
* the model can preserve the voice, which is an important feature towards authentic communication
* the comparison to previous methods is very thorough as many previous methods are reimplemented/compared to

**Weaknesses:**

There is no specific weakness other than most of the novelty comes from the prior work (DA approach, 2-pass decoding framework, fastspeech 2) and this work very effectively combines this prior work to solve a new task.

**Questions:**

* 227: "We implement the following baseline systems." Would you be able to explain this in more details. Does reimplement mean reusing existing results, using existing implementation and retraining models on different datasets, etc.
* 235:  "We remove the R-Drop training [28] for simplification." It would be good to justify this choice, especially if it leads to suboptimal results for the corresponding baseline (and also given that there is an existing implementation for that work).
* 245: "an autoregressive S2TT model." Would it be possible to add details on that S2TT model? Is is a transformer? A particular existing model (if so, add citation, etc.)?


**Limitations:**

The limitations and broader impacts section is very effective in addressing limitations of the work and potential risks.

---

> ### Author Rebuttal · Authors · 2023-08-09
>
> We truly appreciate your positive evaluation of our work. Below, we have provided the responses to each of your valuable comments.
>
> 1. **Re: 227: "We implement the following baseline systems." Would you be able to explain this in more details. Does reimplement mean reusing existing results, using existing implementation and retraining models on different datasets, etc.**
>
> Sorry for the misunderstanding. **"Re-implementation" refers to "using existing implementation and retraining models on different datasets".** Specifically, for S2UT, Translatotron, UnitY, and Translatotron 2, we used the implementations available in the **fairseq** repository without any modifications. For TranSpeech, we utilized the official implementation provided by the authors. Therefore, we can ensure the correctness of our baseline systems' implementations.
>
> Regarding model sizes and training hyperparameters of baseline systems, we strived to match them as closely as possible to the original papers while maintaining consistency among the systems to ensure fairness in the comparisons (See more details in Appendix A). Based on the results of our re-implementations, the performance in translation quality and decoding speed of each system aligns closely with the conclusions reported in the previous papers.
>
> 2. **Re: 235: "We remove the R-Drop training [28] for simplification." It would be good to justify this choice, especially if it leads to suboptimal results for the corresponding baseline (and also given that there is an existing implementation for that work).**
>
> R-Drop is a general regularization method for neural network, which can reduce the inconsistency in model predictions between training and inference by dropout, thereby improving the model's performance. UnitY's training incorporates the R-Drop technique, which has been proven to effectively enhance translation quality. However, this training trick has not been widely adopted by other systems, as seen in recent works like TranSpeech, which did not use it. Using R-Drop training exclusively for UnitY would lead to an unfair comparison with other systems. **We aim for the comparisons between different models to primarily depend on the modeling approach (e.g., 1-pass vs. 2-pass model structures, choosing units vs. mel-spectrograms as prediction targets), rather than being influenced by specific training tricks.** Additionally, R-Drop training involves duplicating the input, which increases the training cost. Therefore, in our experiments, we have removed the R-Drop training from UnitY. In the future, we plan to investigate the effects of R-Drop training on our DASpeech.
>
> 3. **Re: 245: "an autoregressive S2TT model." Would it be possible to add details on that S2TT model? Is it a transformer? A particular existing model (if so, add citation, etc.)?**
>
> Thank you for pointing out this issue. The speech encoder of the autoregressive S2TT model is the same as that used in DASpeech (subsampler + Conformer encoder). The decoder is a conventional 4-layer transformer decoder. We will provide a more detailed description in the final version.

---

> > ### Comment · Reviewer_nvMe · 2023-08-21
> > **thank you for the answers**
> >
> > Thank you for taking the time to reply to my review.
> >
> > Regarding 1. and 2. I would suggest for extra fairness in comparison to prior work to:
> > * if the language pair setup is the same, to not only report the numbers obtained by running an existing implementation on the setup but also to report existing numbers from the literature.
> > * similarly, regarding R-Drop, I would suggest reporting the numbers that use it as well as reporting numbers obtained by using it in the reimplementation/rerunning of experiments.
> >
> > Best

---

> > > ### Author Response · Authors · 2023-08-21
> > > **Thank you for your response and suggestions.**
> > >
> > > Thank you for your response and suggestions. We will carefully consider your suggestions in the final version.

---

### Official Review · Reviewer_47bw · 2023-07-07

**Soundness:** 2 fair
**Presentation:** 2 fair
**Contribution:** 2 fair
**Rating:** 5
**Confidence:** 4

**Summary:**

The paper introduces DASpeech, a directed acyclic transformer model for fast and high-quality speech-to-speech translation (S2ST). DASpeech employs a two-pass architecture, using a linguistic decoder based on DA-Transformer and an acoustic decoder based on FastSpeech 2. The proposed model captures the complex multimodal distribution of the target speech and achieves comparable or better performance than Translatotron 2, with significant speedup compared to autoregressive models. DASpeech also demonstrates the ability to preserve the source speaker's voice during translation.


**Strengths:**

By incorporating the DA-Transformer and FastSpeech 2, DASpeech leverages the strengths of these models to capture both linguistic and acoustic diversity, resulting in high-quality translations.

**Weaknesses:**

1. The novelty is not enough, the method of this article seems to just learn from the DAT method of NMT and use it in ST
2. While the paper addresses the challenges of high decoding latency in S2ST models, it would be beneficial to provide more detailed analysis or experimentation on the trade-off between translation quality and decoding speed in DASpeech.


**Questions:**

Could the authors elaborate on any specific differences between DASpeech and the DA Translation particularly?

---

> ### Author Rebuttal · Authors · 2023-08-09
>
> Thank you for providing your valuable and constructive feedback! Please find below the responses to each comment.
>
> 1. **Re: The novelty is not enough, the method of this article seems to just learn from the DAT method of NMT and use it in ST (Specific differences between DASpeech and the DA-Transformer particularly)**
>
> Thank you for pointing that out. In our opinion, applying the DA-Transformer to **speech-to-text translation** is straightforward as the prediction target remains the target text, and only the text encoder needs to be replaced with a speech encoder. **However, its application to speech-to-speech translation (S2ST) is non-trivial.** For the two-pass S2ST model, predicting the target text is just the first step; it also requires an acoustic decoder to further synthesize the target speech based on the output hidden states from the linguistic decoder. During the training process, **we need to find the decoder's hidden state sequence corresponding to the ground truth target text.** For autoregressive models like UnitY and Translatotron 2, teacher forcing training naturally addresses this problem. However, for the DA-Transformer, which utilizes a Directed Acyclic Graph (DAG) to model multiple possible translations simultaneously, the decoder's output consists of representations for all nodes in the DAG. To find the hidden states corresponding to the ground truth text, we proposed the **"expect-path training"** method (Section 3.2). This method considers all possible paths in the DAG and calculates the posterior probabilities $P_\theta(a_i|X,Y)$ through dynamic programming. Based on these posterior probabilities, we are able to compute the **expected hidden states** corresponding to each word in the ground truth text. This enables us to jointly train the DA-Transformer and the Text-to-Speech (TTS) model for the S2ST task.
>
> **As Reviewers nvMe and WJ3s mentioned, while the novelty of our paper builds upon prior works, effectively integrating them to address a new and challenging task is also considered novel. Note that our approach is not limited to a specific TTS model**. On the contrary, our focus is on efficiently integrating the S2TT DA-Transformer and the TTS model, enabling end-to-end training for the S2ST task. As a result, theoretically, our method can also combine the S2TT DA-Transformer with other TTS models that are more powerful than FastSpeech 2, allowing for the modeling of more diverse and complex target speech. We leave this for future research. We hope that our work can serve as a strong baseline for non-autoregressive speech-to-speech translation in the future.
>
> 2. **Re: More detailed analysis or experimentation on the trade-off between translation quality and decoding speed in DASpeech.**
>
> Thank you for your advice. Following your suggestion, we have included several additional experiments concerning decoding efficiency of DASpeech. These experiments provide further insights into the trade-off between translation quality and decoding speed of DASpeech. **You can refer to the "Global Response" for more details.**

---

### Author Rebuttal · Authors · 2023-08-09

We sincerely appreciate the valuable comments from all the reviewers, which have provided us with many insightful suggestions. **We are greatly encouraged by the fact that most reviewers have found our work to be sound and effective, with a novel application in the speech-to-speech translation (S2ST) task.** To address the concerns raised by the reviewers, we have added some new experiments and responded individually to the comments of each reviewer. We greatly look forward to your feedback. **Should you have any further questions, please feel free to continue sharing your comments.** We are enthusiastic about continuing to address your inquiries during the reviewer-author discussion period.

### **New Experiments Related to the Decoding Speed**

Specifically, following the suggestions of Reviewers 47bw and RzGP, **we have conducted several additional analysis experiments related to the decoding speed of DASpeech.** These experiments include:
1. **Trade-off between translation quality and decoding speed (Please refer to Figure 1 in the additional PDF provided in our "Global Response"):** By adjusting the hyperparameter $\lambda$ of DASpeech, the translation quality and decoding latency will change. Specifically, as $\lambda$ increases, the decoding latency of the model will increase, and it achieves the best translation quality when $\lambda=0.5$. Compared to previous autoregressive models (e.g. UnitY and Translatotron 2), DASpeech achieves a significantly higher speedup while maintaining comparable translation quality. When compared to the previous non-autoregressive model TranSpeech (which can balance translation quality and decoding latency by adjusting the number of iterations), DASpeech shows significant improvements in both translation quality and decoding speed. In conclusion, compared to existing works, **DASpeech achieves the best trade-off between translation quality and decoding latency.**
2. **Translation latency of different models across different source speech lengths (Please refer to Figure 2 in the additional PDF provided in our "Global Response"):** We have conducted a study on the translation latency of different models for speech inputs of different lengths. The results indicate that the translation latency of autoregressive models (S2UT, Translatotron, UnitY, Translatotron 2) significantly increases with the length of the source speech. On the other hand, the translation latency of non-autoregressive models (TranSpeech, DASpeech) are hardly affected by the source speech length. **When translating longer speech inputs, DASpeech's decoding speed can reach more than 20 times that of S2UT.**
3. **Speedup ratio of DASpeech under batch decoding (Please refer to Figure 5 in Appendix D provided in our original submitted supplementary materials):** Previous work pointed out that the speed benefits of non-autoregressive models may degrade during batch decoding. To better understand this problem, we evaluate the speedup ratio under different decoding batch sizes. **We find that DASpeech ($\lambda=0.5$ with Joint-Viterbi decoding) still achieves more than 6x speedup with a decoding batch size of 64.** Please refer to Appendix D for further details and information on this analysis.

### **New Experiments on the CVSS-C De-En Dataset**

In addition, based on the suggestions from Reviewer PNQE and WJ3s, **we conducted additional experiments on the CVSS-C De-En dataset.** This language direction poses greater difficulty due to significant word order differences between English and German, compounded by a relatively small training set of only 127K data. The ASR-BLEU scores on the CVSS-C De-En test set are presented in the table below. We observed that, possibly due to the aforementioned challenges, single-pass models (S2UT, Translatotron, TranSpeech) performed poorly in this language direction, whereas 2-pass models showed obviously better performance. Notably, **our DASpeech outperformed the state-of-the-art autoregressive models in the De-En direction**, confirming the effectiveness of DASpeech.



| Models                                    | ASR-BLEU (CVSS-C De-En) |
| ---------------------------------------- | ----- |
| S2UT                                     | 2.99  |
| Translatotron                            | 1.97  |
| UnitY                                    | 15.33 |
| Translatotron 2                          | 15.80 |
| TranSpeech (iter=5)                      | 1.90  |
| TranSpeech (iter=15, b=15, NPD)          | 2.28  |
| DASpeech ($\lambda=0.5$, Lookahead)      | 15.97 |
| DASpeech ($\lambda=0.5$, Joint-Viterbi) | **16.14** |

---

### Decision · Program_Chairs · 2023-09-21

**Decision:**

Accept (poster)

**Comment:**

Most of the reviewers felt that the results looked decent. A couple of questions were raised about testing on more languages by PNQE and WJ3s, which the authors offerred during the rebuttal and the new results seemed to enforce the quality of the results (at least according to reviewer WJ3s). The authors were asked to validate their claim about SOTA results on some of the data by RzGP, which I think the authors clarified, appropriately.

47bw, RzGP raised their opinion that the method was not novel and included stacking two prior works. The authors rebuttal, emphasized the novelty being in -- the "expect-path training" method (Section 3.2). To me that seems like a fair argument, and actually quite novel -- the DA Transformer uses alignments for translation, but this method goes one step further and uses the uncertainty in the transition matrix for translation and manages to connect it to the speech generation part quite effectively.